# Seismic monitoring of the Auckland Volcanic Field during New Zealand's COVID-19 lockdown

Kasper van Wijk[1], Calum J. Chamberlain[2], Thomas Lecocq[3], and Koen Van Noten[3]

[1]Department of Physics, University of Auckland
[2]School of Geography, Environment and Earth Sciences, Victoria University of Wellington
[3]Royal Observatory of Belgium,Seismology-Gravimetry

**Correspondence:** Kasper van Wijk (k.vanwijk@auckland.ac.nz)

**Abstract.**

The city of Auckland, New Zealand (Tāmaki Makaurau, Aotearoa) sits on top of an active volcanic field. Seismic stations in and around the city monitor activity of the Auckland Volcanic Field (AVF), and provide data to image its subsurface. The seismic sensors – some positioned at the surface and others in boreholes – are generally noisier during the day than the night. For most stations weekdays are noisier than weekends, proving human activity contributes to recordings of seismic noise, even on seismographs as deep as 384 m below the surface, and as far as 15 km from Auckland's Central Business District. Lockdown measures in New Zealand to battle the spread of COVID-19 allow us to separate sources of seismic energy and evaluate both the quality of the monitoring network, as well as the level of local seismicity. A matched-filtering scheme based on template matching with known earthquakes improved the existing catalogue of 5 known local earthquakes to 35 for the period between November 1st, 2019 and June 15th, 2020. However, the Level 4 lockdown from March 25th to April 27th – with its drop in anthropogenic seismic noise above 1 Hz – did not mark an enhanced detection level. Nevertheless, it may be that wind and ocean swell mask the presence of weak local seismicity, particularly near surface-mounted seismographs in the Hauraki Gulf that show much higher levels of noise than the rest of the local network.

## 1 Introduction

The Auckland Volcanic Field (AVF) is an active intra-plate volcanic field consisting of 53 known volcanoes (Hopkins et al., 2020). The last eruption ∼600 years ago is responsible for Rangitoto Island, a prominent geologic feature in the Hauraki Gulf, ∼10 kilometers from the Auckland Central Business District (CBD). Given the risk involved for the 1.5 million inhabitants of a city built on top of the AVF, a seismic network (AVSN, Figure 1 and Table 1) monitors for early warning signs of an impending eruption (Sherburn et al., 2007). However, seismic recordings in urban environments suffer from contamination by anthropogenic noise. To minimise the recording of anthropogenic noise in Auckland, seven of ten stations of the AVSN are installed in boreholes. This measure also reduces the recording of seismic signal from wind and ocean waves (see Table 1 and

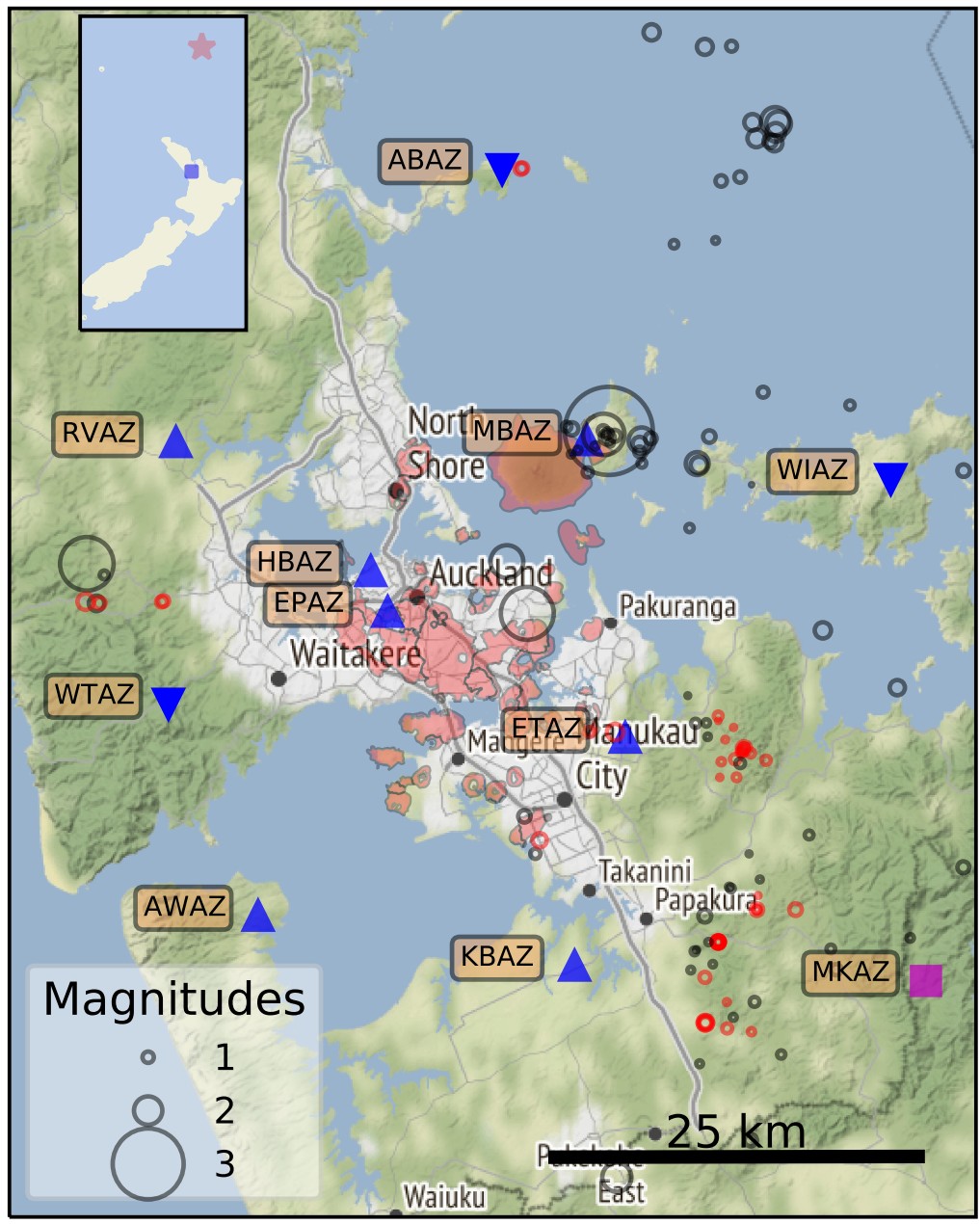

**Figure 1.** Terrain map of the greater Auckland area from OpenStreetMap with surface expressions of Auckland's volcanoes in pink from the Determining Volcanic Risk in Auckland (DEVORA) project. The stations of the Auckland Volcanic Seismic Network (AVSN, managed by Geonet) are presented as blue triangles (surface stations triangles point upwards; boreholes point downwards). Station MKAZ (magenta square) is a broadband surface seismic station from the national seismic network. Earthquake epicentres (black circles) and quarry blasts (red circles) are from the Geonet catalogue from January 1st, 2011, to June 15th, 2020. The inset shows the position of Auckland in New Zealand, as well as the epicentre of an earthquake in the Kermadec Islands region (event ID us60008fl8).

**Table 1.** Coordinates of the AVSN short-period seismometers, and the broadband station MKAZ.

| Station | Latitude | Longitude | Depth (m) |
|---------|----------|-----------|-----------|
| ABAZ | -36.600 | 174.832 | 0 |
| AWAZ | -37.064 | 174.643 | 371 |
| EPAZ | -36.875 | 174.744 | 383 |
| ETAZ | -36.953 | 174.928 | 347 |
| HBAZ | -36.850 | 174.730 | 380 |
| KBAZ | -37.095 | 174.889 | 160 |
| MBAZ | -36.769 | 174.898 | 93 |
| MKAZ* | -37.10413 | 175.16117 | 0 |
| RVAZ | -36.770 | 174.579 | 250 |
| WIAZ | -36.793 | 175.134 | 0 |
| WTAZ | -36.932 | 174.573 | 0 |

Ashenden et al., 2011; Boese et al., 2015). The seismic data from the AVSN are hosted by GeoNet (https://www.geonet.org.nz/), and publicly available in near-real time.

New Zealand entered a Level 4 lockdown to combat the spread of COVID-19 at 23.59pm on the 25th of March, 2020 (local time). Schools were closed, work was halted or moved to home, and travel reduced to trips to the doctor and the supermarket. A limited workforce continued to work and commute if their profession was deemed "essential." On the 27th of April, New Zealand lowered this lockdown to Level 3, which meant mobility of Aucklanders increased, and construction work, for example, resumed. The following results present the impact of the lockdown on seismic recordings of the AVSN. Recent

studies have shown impact in the 1-10 Hz (Poli et al., 2020) and 4-14 Hz (Lecocq et al., 2020) range. The AVF is an active volcanic field with ongoing efforts to image the subsurface (Ensing et al., 2017; Ensing and van Wijk, 2018; Ensing, 2020) with seismic data that spans the entire seismic data spectrum. Therefore, our analysis includes the impact of the lockdown on seismic data from 0.1 to 50 Hz. In addition, we use this uniquely quiet period of Auckland in lockdown to 1) evaluate the sensitivity of the AVSN and 2) explore the seismic character of the Auckland Volcanic Field by increasing the earthquake

catalogue with a matched-filtering technique with known template earthquakes.

## 2   Results

Figure 2 displays 24 hours of the vertical component of ground velocity measured on station HBAZ, a short-period seismometer installed in a borehole, 380 m underground in Herne Bay, Auckland. The dominant feature in this seismogram is signal at 10:06UTC from a M6.4 earthquake in the Kermadec Islands region (event ID us60008fl8). In general, however, the seismogram

is less noisy during the (local) nighttime than during the daytime. As a result, signals associated with the smallest earthquakes

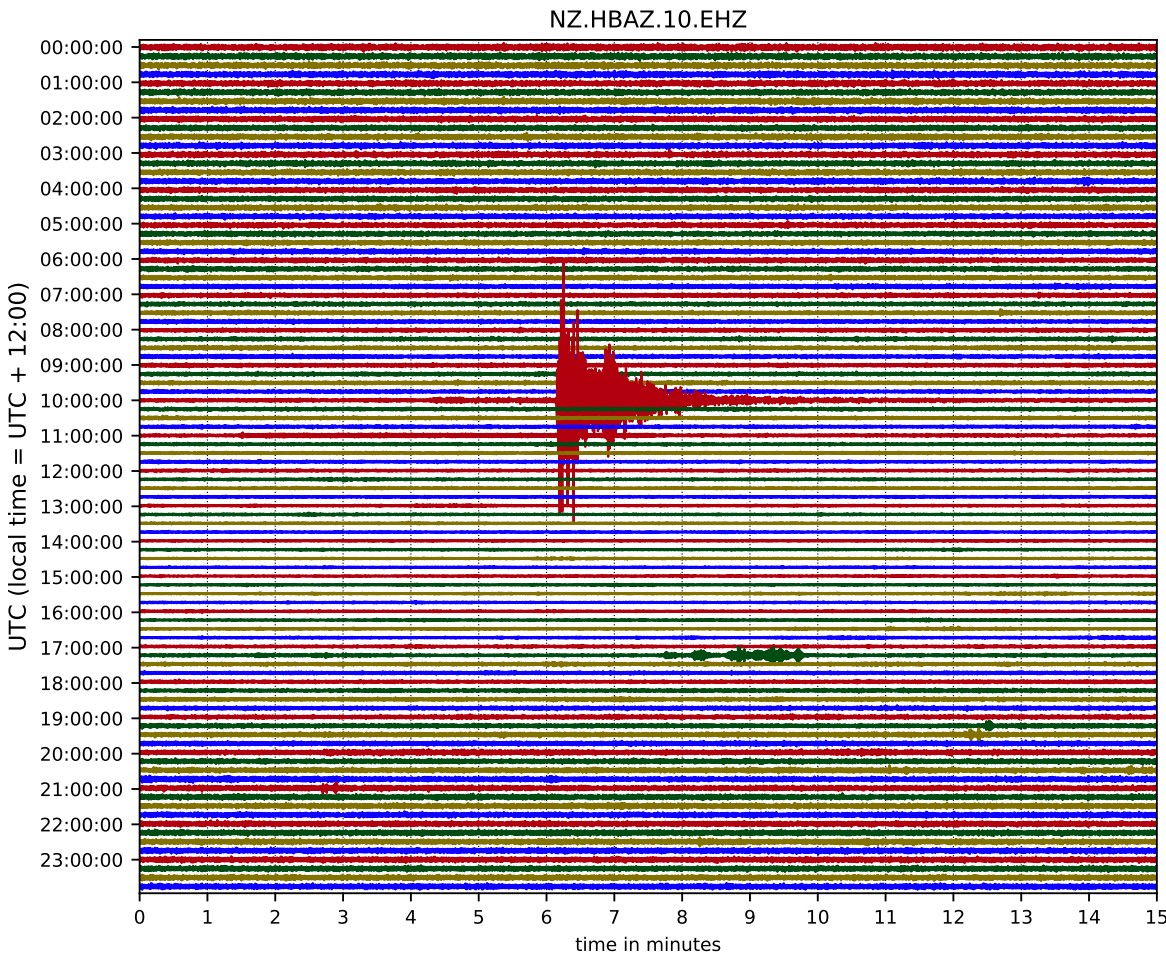

**Figure 2.** Vertical component of the seismic wavefield at borehole station HBAZ for March 14th, 2020. The signal with the largest amplitudes is from an earthquake in the Kermadec Islands region (event ID us60008fl8).

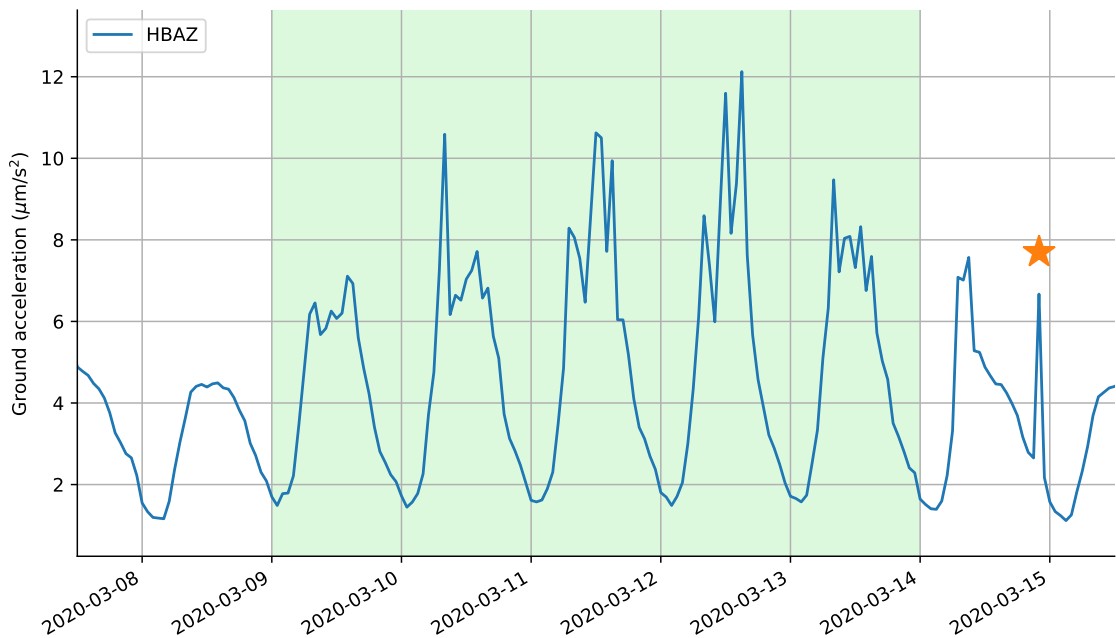

**Figure 3.** Standard deviation over 30-minute intervals of the ground acceleration of short-period borehole station HBAZ. Times are in local NZT, and weekdays have a green background. Note the lower night-time noise level compared to day-time, and quieter weekends. The origin time of the Kermadec Regions event ID us60008fl8 is annotated with a star.

detectable at night could be masked by noise during the day. An example is the small event around 17.23(UTC, in green), which may have been obscured by day-time noise levels.

To study seismic signal levels over longer time periods, we compute the standard deviation in 30-minute time windows, after an instrument correction to acceleration, as defined in the volcano monitoring technique called Real-Time Seismic Amplitude
Measurement (RSAM, Endo and Murray, 1991). We filter the data between 0.1 Hz and 50 Hz to both cover the range of frequencies of interest in volcano monitoring and in seismic tomography. Figure 3 displays the standard deviation from the 8th to the 15th of March (2020) on station HBAZ in local time. From here on, all time scales are in the local time zone and week-days are marked by a light green background. In addition to a difference between day- and nighttime noise, there is a clear distinction between weekdays and weekends: especially data on Sundays appear less noisy. RSAM for HBAZ varies
between 6 and 12 $\mu$m/s$^2$ during the day on weekdays, while the night-time RSAM values are less than 2 $\mu$m/s$^2$. The narrow spike late on the March 14th is due to the previously mentioned earthquake in the Kermadec Islands region.

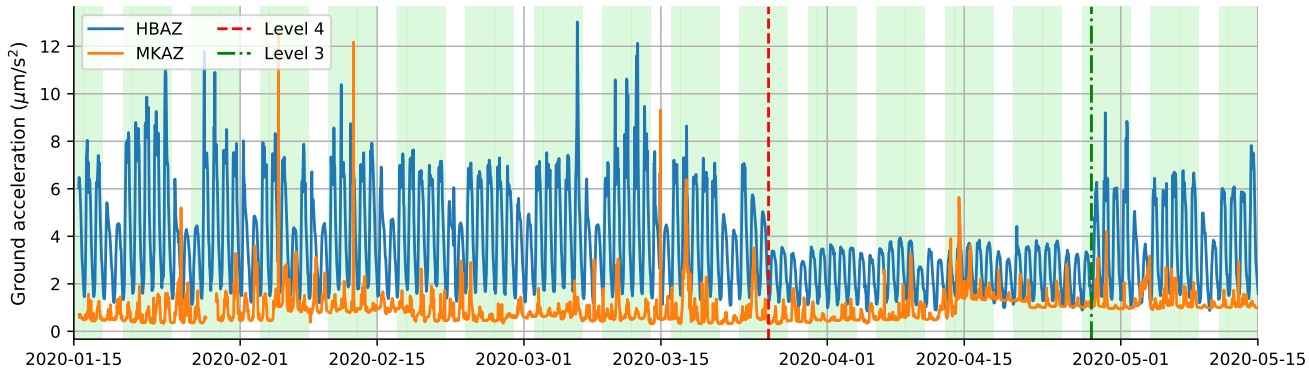

**Figure 4.** RSAM at short-period borehole seismometer HBAZ and the nearest broad-band station MKAZ. In addition to the contrasts in RSAM values for days, nights, weekends and weekdays, the Level 4 lockdown period (its start marked by a red dashed line and the transition to Level 3 is annotated by a green dash-dotted line) is marked by lower RSAM values for station HBAZ.

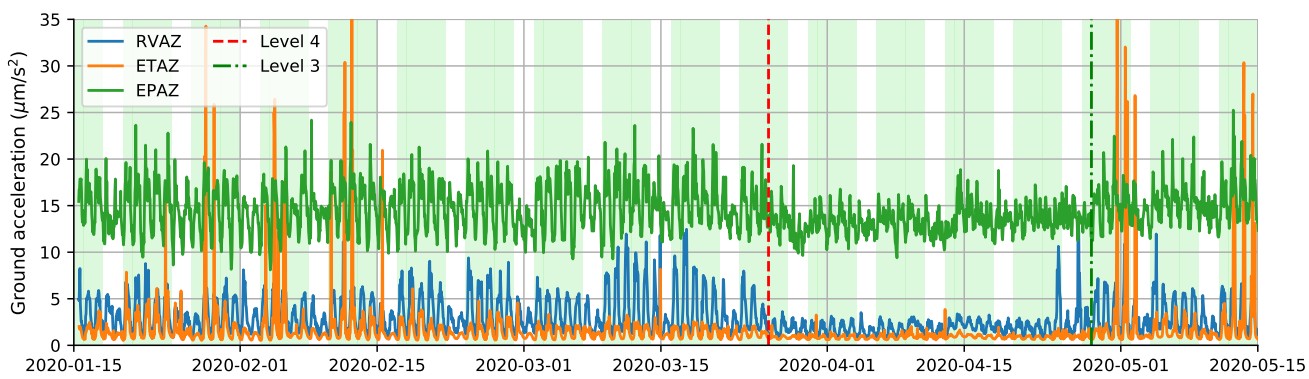

**Figure 5.** RSAM levels for three other borehole stations of the AVSN nearest to the Auckland CBD are reduced during the lockdown.

## 2.1 Seismic noise levels during a Level 4 lockdown

Restrictions during a level-4 lockdown to combat the COVID-19 pandemic in New Zealand resulted in a reduction of weekday day-time RSAM values on HBAZ (Figure 4). Before the lockdown, indicated by the red vertical dashed line, the periodicity of
55 RSAM follows the familiar day/night and weekend/weekday pattern from Figure 3, but after the Level 4 lockdown ended (at the dash-dotted green vertical line) weekday daytime RSAM levels resemble those of a typical Sunday. For comparison, the nearest broadband station south of Auckland, MKAZ, generally has lower RSAM levels and appears unaffected by the New Zealand lockdown.

The lockdown measures affect AVSN data in different ways. Stations EPAZ, RVAZ and ETAZ are – as station HBAZ – in a
60 borehole near the CBD. And similarly to station HBAZ, their RSAM values captured in the top panel of Figure 5 are reduced

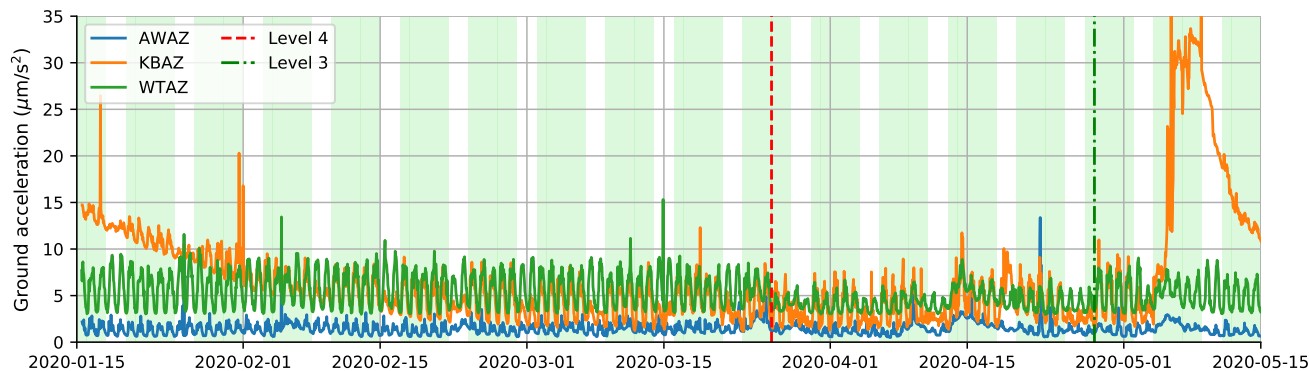

**Figure 6.** RSAM levels at three AVSN stations Southwest of Auckland's CBD. Only RSAM levels on station WTAZ dropped during the lockdown.

at the start of the Level 4 lockdown and increased to pre-lockdown levels after the severest restrictions were lifted in Level 3. In addition, station EPAZ located under Eden Park Stadium suffers from a continuous source of high-frequency (∼50 Hz) noise, which results in elevated RSAM values at all times. Figure 6 contains RSAM values for stations to the Southwest of the Auckland CBD. WTAZ data are less noisy during the lockdown, despite not showing a significant weekday/weekend signature. Conversely, KBAZ has quieter weekends, but no reduction in RSAM during the lockdown. Furthermore, the data for KBAZ are marked by extended periods of larger RSAM values in the beginning of January and mid-May. Even though KBAZ and WTAZ are borehole stations, the least noisy station of these three is station AWAZ, located in a borehole on the Awhitu Peninsula. The data from this station do not show the weekend/weekday signature, nor a lockdown reduction in RSAM values.

### 2.1.1 The influence of wind

Figure 7 presents the RSAM values for three seismic stations in the Hauraki Gulf. The two seismic surface stations ABAZ and WIAZ are approximately 25 times noisier than the other stations in the AVSN. To compare the features in the RSAM data for all three stations, the MBAZ signal is multiplied by a factor of 25. Station MBAZ is in a borehole on Motutapu Island in the Hauraki Gulf; a seismically quiet location. There are no distinctions between weekdays and weekends, but still a small reduction in noise levels during the lockdown is evident in the data. Figure 7 includes the running average of wind speed over a 72-hour window from Auckland's Sky Tower in the CBD. Correlation between the noisiest periods and the wind is strongest for the surface stations, for example during high-wind times in the middle of February and April, as well as in early May.

### 2.1.2 Mobility data

Figure 8 contains the same RSAM values for HBAZ as in Figure 4, but filtered between 1-14 Hz. This frequency band has previously proved to be most sensitive to human activity (Dias et al., 2020), but is also in the range of frequencies of interest for seismic monitoring of volcanic unrest. Figure A1 shows that RSAM values drop for even higher frequencies, possibly as

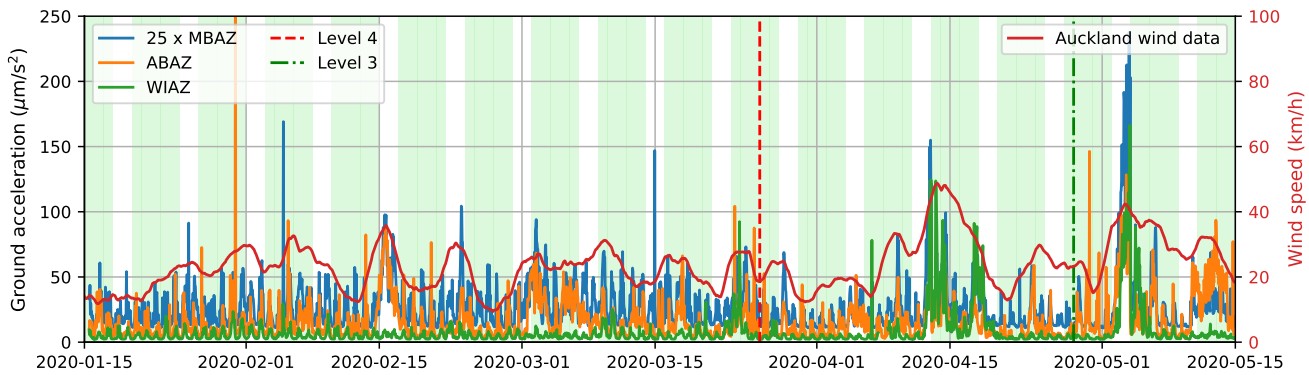

**Figure 7.** RSAM values for stations east of Auckland CBD, on the Hauraki Gulf. Only the borehole RSAM data from MBAZ, artificially multiplied by 25 for visual purposes, correlate with the Level 4 lockdown in New Zealand.

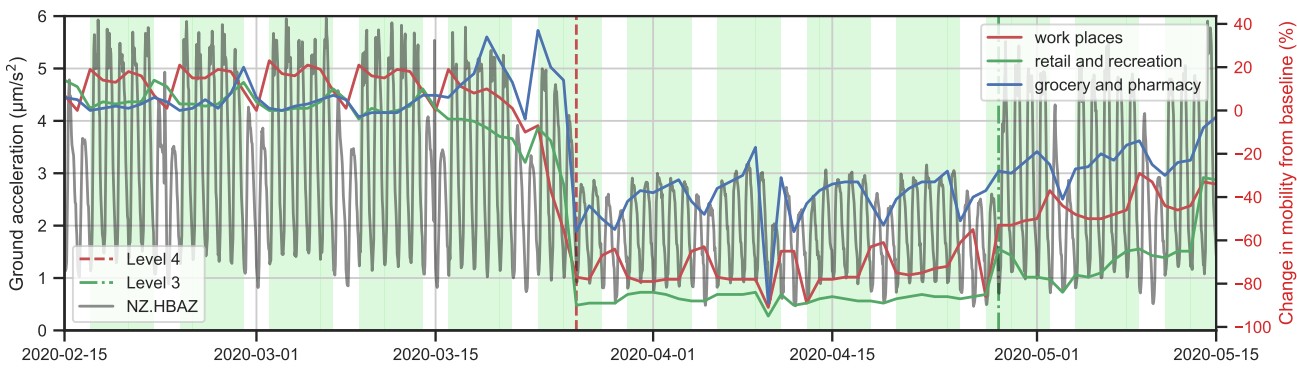

**Figure 8.** RSAM values for station HBAZ, compared to Google's mobility data for Auckland.

high as the Nyquist frequency in the data (50 Hz). The right vertical axis is for Google's public mobility data for Auckland (last accessed June 2020). The mobility data are broken down into different human activities. Prior to the lockdown, the strongest correlation between mobility data and seismic noise levels is seen in the Work Places category; in both cases, the weekend leads to a drop in noise and work-place activity. Not surprisingly, all but the residential activity (movement inside the home) dropped significantly during the lockdown, but the correlation between seismic noise on HBAZ and mobility associated with "grocery and pharmacy" is particularly strong. Because the mobility data are presented as a change in activity, the workplace activities dropped overall, but work deemed essential continued at the weekends, resulting in a temporary increase in weekend workplace activity during the lockdown period.

### 2.1.3   Earthquake detection

To test whether the reduction in noise during lockdown affects the detectability of local seismicity, we employed a network matched-filter detector (Gibbons and Ringdal, 2006) to construct a catalogue of the AVF. The matched-filter method complements standard energy-based detection methods that rely on variations in waveform amplitude and are therefore strongly controlled by background noise-levels. Matched-filters commonly provide robust earthquake detections at lower amplitudes (and therefore magnitudes) than standard detectors, often generating catalogues with one magnitude unit lower completeness (around 10 times more earthquake detections Warren-Smith et al., 2018; Shelly and Hardebeck, 2019; Ross et al., 2019). All codes to generate the following results can be retrieved from Chamberlain and van Wijk (2020).

We made earthquake templates from the Geonet catalogue with 59 events between 1st January 2011 and 15th April 2020 that had picks on at-least five stations of the AVSN. Templates are constructed by re-sampling the seismic data to 50 Hz and filtered between 2–15 Hz using a fourth-order Butterworth band-pass filter. Template waveforms were cut to 6 s length starting 0.5 s before P-wave picks on vertical channels and 0.5 s before S-wave picks on horizontal channels. We required a minimum signal-to-noise ratio of 5 to retain waveforms in the templates. All 59 templates were correlated with data processed with the same parameters between the 1st of November 2019 and the 15th of June 2020 using EQcorrscan (Chamberlain et al., 2018). Detections are made when the network sum of the normalised cross-correlations exceeds 9 times the median absolute deviation (MAD) of the cross-correlation sum for that day (Warren-Smith et al., 2017). If more than one detection occurs within 2 s of another detection, the detection with the highest average correlation value is retained.

To improve the quality of the resulting catalogue, we computed cross-correlation derived pick-corrections for each detection using the method (Warren-Smith et al., 2017) with a minimum normalised cross-correlation value of 0.4 required for each pick. We retained detections that were picked on at least three stations, resulting in a final catalogue of 40 events. One of these events is a quarry blast, and a further four events are not related to visible phase-arrivals. In the same time-period in which we detect 35 earthquakes, the GeoNet catalogue contains five earthquakes (which are a subset of the events in Figure 9).

## 3   Discussion

Seven of the ten stations of the AVSN are installed in boreholes, reducing the impact of environmental and anthropogenic noise sources. For example, station AWAZ at 371 m below the surface is inline with the runways of the Auckland airport, and appears insensitive to airplane noise, compared to surface station WTAZ. Nevertheless, borehole stations closest to the CBD remain sensitive to anthropogenic noise, which was reduced during the level 4 lockdown.

Correlation of RSAM values during the lockdown at station HBAZ are strongest with the "grocery and pharmacy" category of Google mobility data. This station is 380 m underground, but the local grocery and medical centre are less than 2 km away.

Surface stations on Waiheke Island (WIAZ) and on the Whangaparaoa peninsula (ABAZ) are the noisiest stations, and RSAM values correlate with wind speed. In fact, the wind data are the result of a moving average of 72 hours of data during which winds vary significantly. Therefore, we believe the correlation between seismic noise and this averaged wind speed indicates the correlation is more related to the ocean swells associated with storms. In fact, we believe all the seismic stations

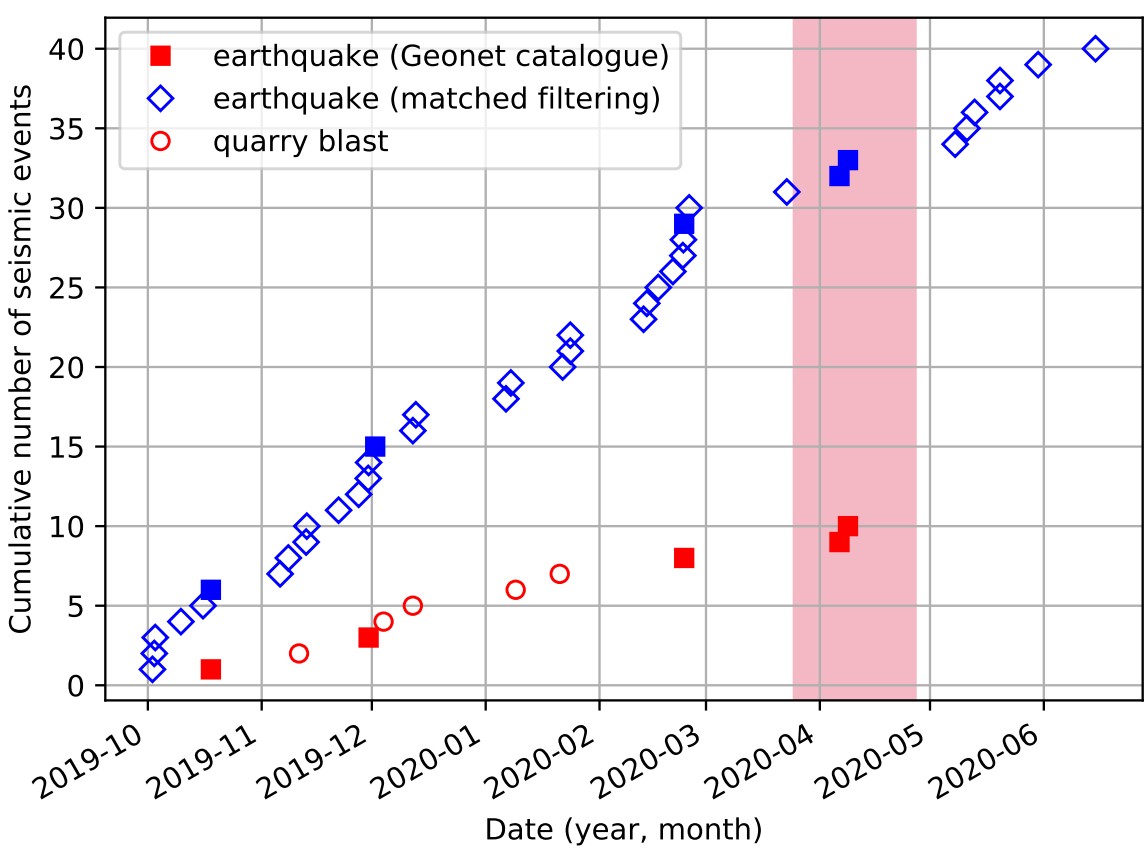

**Figure 9.** Matched filtering with templates built from Geonet-catalogued earthquakes results in 40 events between October 1st, 2019, and June 15th, 2020. Of these 40 events, 35 are identified as local earthquakes, including the five earthquakes in the Geonet catalogue for the same period.

of the AVSN are sensitive to ocean swell, indicated by variations in RSAM amplitudes between day and night. Installing WIAZ and ABAZ in boreholes would reduce their noise floor by one or two orders of magnitude, based on RSAM values on nearby borehole station MBAZ and others, improving sensitivity of the AVSN network for both volcano monitoring and seismic tomography.

Large amplitudes in RSAM for station KBAZ in January and May 2020 cannot be linked to storms or lockdown effects, and may be the result of local farming activity. Additional variations in weekday/weekend RSAM signals are robust even in the presence of the lockdown. We attribute this to the proximity of one of the main highways that carry transport of (essential) services in and out of Auckland.

The reduction in anthropogenic noise is noted in the frequency band from 1 Hz and up, possibly all the way to the Nyquist frequency in the data, 50 Hz (see Figure A1 in Appendix A). Anthropogenic noise in this frequency band affects both volcano monitoring with RSAM, as well as its ability to detect weak and local seismicity. Nevertheless, the reduction in seismic noise throughout New Zealand's level 4 lockdown does not appear to have affected our ability to detect earthquakes in this seismically quiet part of New Zealand. By employing a matched-filter approach, we limit the sensitivity of our detector to variations in noise amplitude. Analysis of the cross-correlation sums for the full period studied shows little variation in this detection statistic (see Figure B1 in Appendix B), resulting (combined with low seismicity rates) in little change in detection rate. In contrast, classic earthquake detectors dependent of ratios of seismic amplitudes, would be sensitive to this variation in noise. This highlights the efficacy of matched-filter detectors for consistent detection capability during periods of variable noise.

## 4   Conclusions

During the Level 4 lockdown to combat the COVID-19 pandemic in New Zealand, six of the ten seismometers that monitor the Auckland Volcanic Field display reduced seismic noise levels above 1 Hz. One seismic station has low levels of noise all the time (AWAZ), and one station (KBAZ) appears to be affected by local farming and a motorway supporting essential services. Two other (surface) stations (WIAZ and ABAZ) in the Hauraki Gulf have Real-Time Seismic Amplitude Measurement (RSAM) values that are one to two orders of magnitude greater than the rest of the AVSN, and show the strongest correlation with ocean swell.

Besides analysing the sensitivity of the network, the lockdown period allowed us to explore for local earthquakes that are smaller than normally would be detectable. A first attempt at template matching resulted in the detection of 30 new local events in the time where the Geonet catalogue contains only five local earthquakes. However, the detection rate was not higher during the lockdown period than in the periods before or after the lockdown. More advanced matched-filtering detection efforts with existing data are underway, but re-installing surface stations WIAZ and ABAZ in boreholes can improve seismic monitoring and tomography of the AVF.

*Code availability.* All codes for the event detections are available from Chamberlain and van Wijk (2020)

*Author contributions.* All listed authors wrote the manuscript. KvW did the seismic noise analysis. The original idea to analyse seismic data
during the COVID-19 pandemic is from TL and KVN, EQcorrscan analysis by CJC

*Competing interests.* We note no competing interests

## 5   Acknowledgements

Seismic data for the AVSN and MKAZ were made available through the GeoNet project, sponsored by the New Zealand
Government through its agencies: Earthquake Commission (EQC), GNS Science and Land Information New Zealand (LINZ).
The wind speed data were made available by NIWA. Data processing was done in ObsPy (Krischer et al., 2015), and data
visualisation with Matplotlib (Hunter, 2007) and Cartopy (Met Office, 2010 - 2015) in Python.

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

## Appendix A: Spectrograms

Figure A1 contains the spectrograms for every station of the AVSN, plus the broadband station MKAZ of the national seismic monitoring network. The spectrograms are built using the continuous seismic records. The data is split in windows of 30 minute duration that overlap by 50%. The PSD of each window is calculated using MSNoise (Lecocq et al., 2014),which relies on the Probabilistic Power Spectral Densities (PPSD) implementation of ObsPy (Krischer et al., 2015). the frequency binning is by 1.25% of an octave (default 12.5%, or 1/8) and the PSDs are smoothed over 2.5% of an octave (default 100%, or 1 octave) around the central frequency of each bin. The amplitudes of the PSDs are binned in 0.25 dB bins.

All the stations have a distinct spectral signature near (but just above) 0.1 Hz for event ID us60008fl8. The period between the black vertical dashed lines marks the extent of the lockdown in New Zealand. For most stations, anthropogenic noise is reduced for frequencies from 1 Hz to the Nyquist frequency of 50 Hz. However, the effect of the lockdown is not equally clear in all stations: wind noise on stations on the Hauraki Gulf (WIAZ and ABAZ) dominates over any anthropogenic noise, as previously seen in the seismograms.

## Appendix B: Matched filtering details

The lockdown period is short, especially when compared to seismogenesis in this relatively low seismicity region. Because our template analysis focuses on the efficacy of the matched-filter method, the detectability is affected by the noise-level in the cross-correlation sum. To demonstrate why we do not expect a change in detection-rate during lockdown, we computed and plotted the network cross-correlation sum for one template (GeoNet publicID: 3469372) between February 29th 2020 and May 8th 2020, alongside the amplitude spectra for this time-series (Figure B1. Plotting the full sample-rate correlation-sum shows little power outside the 2-15 Hz range used, however computing the hourly mean correlation-sum provides more useful information regarding the variability in noise in the correlation sum. In this hourly correlation sum, reductions should correspond to reduced noise in the correlation sum and hence enhanced detectability. We find clear daily variations (evidenced by a peak in the amplitude spectra at 24 hour periods), however there is no clear reduction in background correlation values during lockdown. It is based on this evidence that we can be confident that there is no significant change in detectability during lockdown. Note also that our detection threshold is based on the daily median absolute deviation of the correlation sum which further smooths the daily variability in the correlation sum. The range of daily median absolute deviations upon which our threshold is based range from 0.234 to 0.254, with the lowest values falling outside the lockdown period.

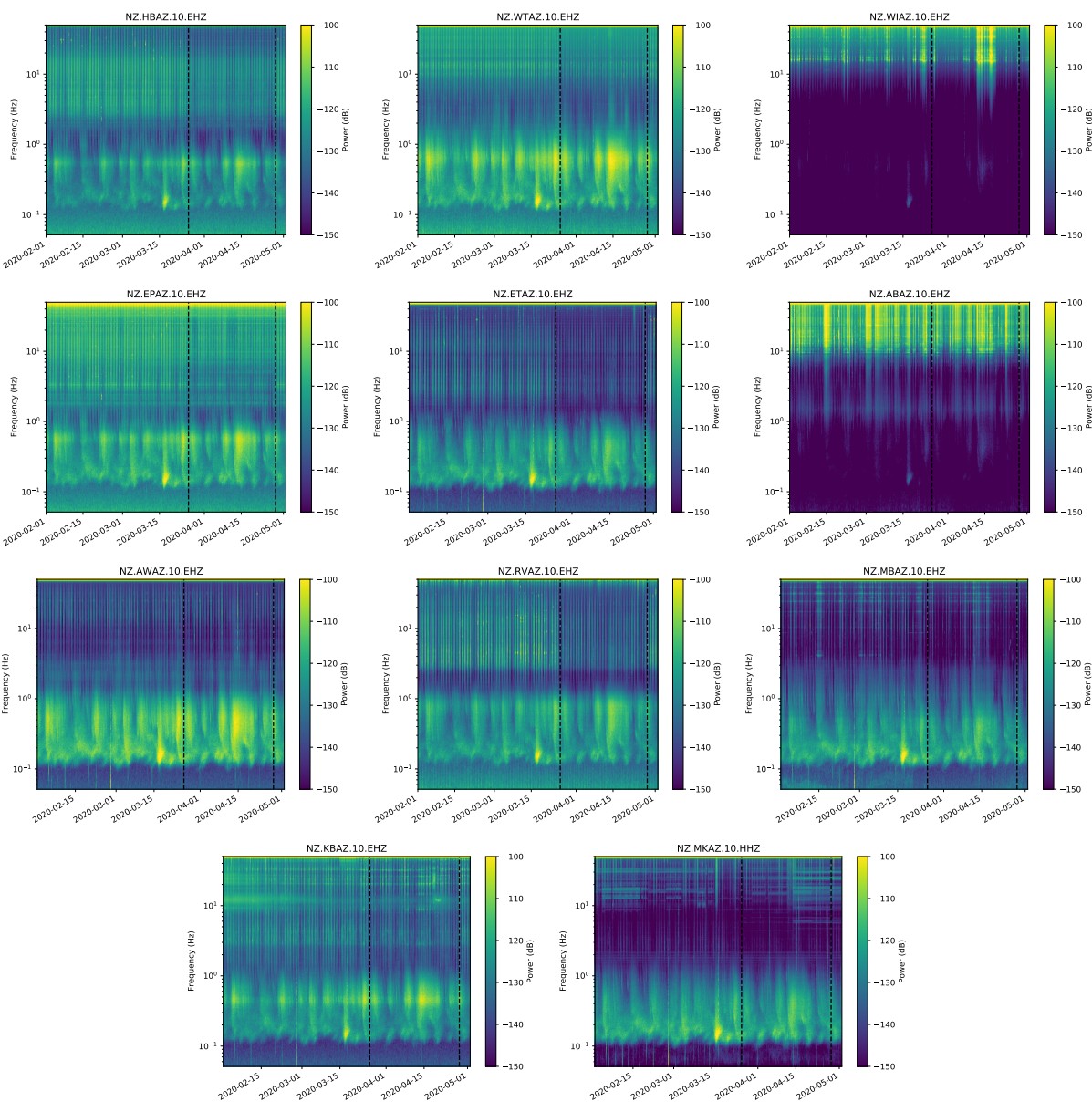

**Figure A1.** Spectrograms for the seismic data of the Auckland Volcanic Seismic Network. The vertical dashed lines indicate the start and end date of the COVID-19 lockdown in New Zealand.

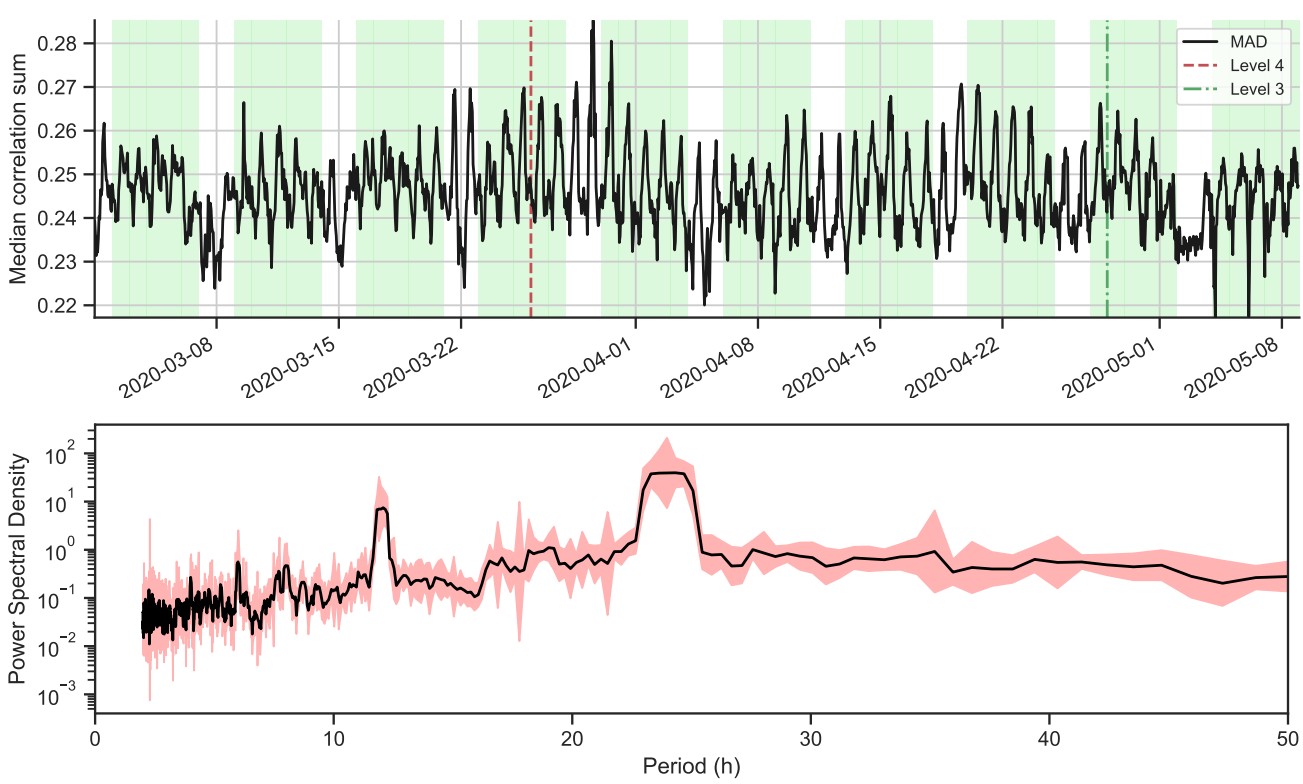

**Figure B1.** Top: the median crosscorrelation sum for the AVSN against a template of one of the Geonet-located earthquakes, showing no reduction in this value for the lockdown period. Bottom: weighted multitaper spectrum with 5 and 95 percent confidence intervals of the top panel time series. Spikes at 12 and 24 hours confirm the noise is dominated by the difference between night and day noise levels.