# Peer review of "Seismic monitoring of the Auckland Volcanic Field during New Zealand's COVID-19 lockdown"

_Solid Earth, 2020_

## Referee Comment (RC1) · Anonymous Referee #1 · 27 Oct 2020

This manuscript "Seismic monitoring of the Auckland Volcanic Field during New Zealand's COVID-19 lock-down" by van Wijk and others analyzed continues seismic data from an array of seismic stations at Auckland Volcanic Field, New Zealand to explore changes in the level of ambient seismic noise related to COVID-19 lockdowns. They also evaluated temporal correlations between seismic noise and environmental & mobility data and the detectability of micro earthquakes during the COVID-19 lockdown period.

Overall, the manuscript is written well and provides interesting insights into seismic noise related to human activity. I have some minor comments.

1. Frequency band.

[Figure]

A main frequency band is 0.1-50 Hz, which seems to include a variety of seismic noise including microseisms, local earthquakes, human activity. I wonder if the authors provide several frequency bands (1-10Hz, 5-15 Hz, 10-30 Hz for example) that may allow us to understand the nature of the ambient seismic noise at the seismic stations.

2. Power spectral density plot.

Related to #1, I think it would be informative to include power spectral density (PSD) plots (with monthly to yearly data before the COVID-19 lockdown) for each station, which would provide the baseline of ambient noise level.

3. Mobility data (work places)

I see the work places data show noise reduction over weekend before the level 4 COVID-19 lockdown but it appears that the work places data periodically increase weekend after the level 4? I may have missed but it would be good to have some comments in the revision.

---

## Referee Comment (RC2) · Anonymous Referee #2 · 9 Nov 2020

I find this is a generally well-written manuscript and an interesting look into a unique period of seismic data. I have two main suggestions for improvement that I think would strengthen this study's arguments significantly using the methods that they've already demonstrated.

First, I think that discussion of changes in earthquake detection during COVID lock-downs would be benefitted by the further context of comparison with other changes in anthropogenic seismic noise levels. Rather than only comparing lockdown to non-lockdown data, adding comparisons of night to day and weekend to weekday could give better insight into how anthropogenic noise affects event detection. The lockdown period is short enough (on the scale of earthquake occurrence rates) that I'm not fully convinced by the authors' claim that there was no change in detection rate during that

period, so it would be helpful to back that up through comparison to other low-noise times for which more data exist.

Second, I believe that this study would be helped with further exploration (or at least explanation) into the frequency domain. The authors say that the 0.1-50 Hz range is of interest to volcano monitoring and contains anthropogenic seismic noise, but don't go into further detail and should at the very least provide more background on that choice of range and show a spectrogram for at least one station. Dividing that range into a few smaller ranges and processing them individually would provide more information about the change in the seismic noise environment (e.g. deconvolving effects of changing wind and water vs changing anthropogenic activity), as well strengthen the authors' arguments regarding those noise levels' effects on event detection.

As for smaller technical corrections, the main things I found were: the authors need to ensure that all data in a figure is included in the one key (e.g. figure 7's key does not contain a red line for wind speed, and figure 8 has two keys instead of one), decide whether to use "lock-down" or "lockdown", and ensure that figures are more colorblind-friendly (e.g. not using red and green for the two different lockdown levels).

―――――――――――――――――――

---

## Referee Comment (RC3) · Anonymous Referee #3 · 15 Nov 2020

The manuscript "Seismic monitoring of the Auckland Volcanic Field during New Zealand's COVID-19 lock-down" presented by van Wijk et al., studies the spatio-temporal variability of anthropogenic noise before, during and after one of the most difficult periods in human history during the 21 century: the COVID-19 pandemic. Using both, borehole and surface seismic stations (broadband and short-period), the authors present a very convincing picture of the reduction in anthropogenic ambient noise (RSAM as called in the manuscript) due the lock-down measures to reduce the spread of the virus and how the reduction in noise amplitude could impact geophysical monitoring of an active volcanic field.

In general, the manuscript reads well and the order of ideas and figures is well presented. I would like the authors to edit or re-write to sentence starting in line 29. I would

suggest to include parenthesis for the references, e.g. (Poli et al., 2014) to separate them from the text, otherwise the reading of the whole sentence is very confusing.

Line 44-45: The authors use the frequency band: 0.1 - 50 Hz assuming the cover the range of interest for volcano monitoring and seismic tomography, however, whiting this range, what frequency band is the most affected? It would be interesting to observe the results presented here using/plotting several frequency bands (5 or 6? ) to understand better where the noise amplitudes show the maximum reduction and how they are related to different anthropogenic activities or sources (diffuse, harmonic, transient, etc) or/and natural processes (volcanic, wind, ocean, etc.).

Line 73: instead of multiplying by a factor of 25, why didn't the authors normalized the time series presented in figure 7? If i understand it correctly, the main idea of the figure is to compare the relative differences between the observed amplitudes at 3 different stations with the wind speed and more importantly evaluate their temporal correlations.

The fact that authors found  35 more earthquakes ( a very low number of events) than Geonet during the lockdown, it doesn't mean they appear because of a reduction in anthropogenic noise, rather, they are found because of the use of a template matching algorithm, that is, systematically more efficient in finding earthquakes when compared with traditional human-based methods. Probably you also could find the same amount of events even without lock down measures. authors can add a sentence like this after line 105.

---

## Author Comment (AC1) · 10 Dec 2020

```
Overall, the manuscript is written well and provides
interesting insights into seismic noise related to human
activity.  I have some minor comments.
```

Thank you, reviewer 1.

```
1.  Frequency band.
```

```
A main frequency band is 0.1-50 Hz, which seems to include
a variety of seismic noise including microseisms, local
earthquakes, human activity.  I wonder if the authors pro-
vide several frequency bands (1-10Hz, 5-15 Hz, 10-30 Hz for
```

example) that may allow us to understand the nature of the
ambient seismic noise at the seismic stations.

2.  Power spectral density plot.

Related to #1, I think it would be informative to include power
spectral density (PSD) plots (with monthly to yearly data
before the COVID-19 lockdown) for each station, which would
provide the baseline of ambient noise level.

To address the reviewers comments (echoed by the other two reviewers), we added
PSD plots in an appendix to show that the anthropogenic component of the noise is at
1 Hz and higher. How high is not exactly clear, and probably varies a bit across the
network, but it appears to approach the Nyquist frequency of 50 Hz. Because we are
interested in earthquakes *and* volcano monitoring for the AVF, we analyse the entire
frequency band of our recordings as one.

3.  Mobility data (work places)

I see the work places data show noise reduction over weekend
before the level 4 COVID-19 lockdown but it appears that
the work places data periodically increase weekend after the
level 4?  I may have missed but it would be good to have some
comments in the revision.

Yes, we added an explanation in the revised manuscript. The key is that the mobil-
ity data is presented as a "change" in mobility. Overall, work place activity dropped
significantly during the lockdown, but essential services continued. During the week-
end, essential services make up a large fraction of workplace activity during normal
times, and this is even more so during a lockdown when non-essential services cease
to happen. The text in the new manuscript reads:

*"Because the mobility data is presented as a change in activity, the workplace activities*

*dropped overall, but work deemed essential continues in the weekends, resulting in a temporary increase in weekend workplace activity during the lockdown period."*

[Figure]

**Figure A1.** Spectrograms for the seismic data of the Auckland Volcanic Seismic Network. The vertical dashed lines indicate the start and end date of the COVID-19 lockdown in New Zealand.

**Fig. 1.**

---

## Author Comment (AC2) · 10 Dec 2020

```
I find this is a generally well-written manuscript and an
interesting look into a unique period of seismic data.
```

Thank you, Reviewer 2.

```
I have two main suggestions for improvement that I think would
strengthen this study's arguments significantly using the
methods that they've already demonstrated.  First, I think
that discussion of changes in earthquake detection during
COVID lock- downs would be benefitted by the further context
of comparison with other changes in anthropogenic seismic noise
```

levels. Rather than only comparing lockdown to non-lockdown
data, adding comparisons of night to day and weekend to weekday
could give better insight into how anthropogenic noise affects
event detection.

We agree, and analyse the (anthropogenic) noise by looking at day and night-time
differences as well (see Figure 2, and 3, for example), as well as weekdays versus
weekends.

The lockdown period is short enough (on the scale of earthquake
occurrence rates) that I'm not fully convinced by the authors'
claim that there was no change in detection rate during
that period, so it would be helpful to back that up through
comparison to other low-noise times for which more data
exist.

Indeed, we did not find more earthquakes during the lockdown than in the time before
and after the lockdown, and agree with the reviewer that more can be done to explain
this. Now, whether this is also true for christmas periods, for example, is for a special
issue on christmas-holiday seismology! All jokes aside: because our analysis focuses
on the efficacy of the matched-filter method, the detectability is affected by the noise-
level in the cross-correlation sum. To demonstrate that we would not expect a change
in detection-rate during lockdown we computed and plotted (now included in a new
appendix of our revised paper) the network cross-correlation sum for one template be-
tween February 29th 2020 and May 8th 2020, alongside the multi-tapered power spec-
trum for this time series. Plotting the full sample-rate correlation-sum shows little power
outside the 2-15 Hz range used, however computing the hourly mean correlation-sum
provides more useful information regarding the variability in noise in the correlation
sum. In this hourly correlation sum, reductions should correspond to reduced noise in
the correlation sum and hence enhanced detectability. We find clear daily variations
(evidenced by a peak in the amplitude spectra at 24 hour periods), however there is no

clear reduction in background correlation values during lockdown. It is based on this evidence that we can be confident that there is no significant change in detectability during lockdown. Note also that our detection threshold is based on the daily median absolute deviation of the correlation sum which further smooths the daily variability in the correlation sum. The range of daily median absolute deviations upon which our threshold is based range from 0.234-0.254, with the lowest values falling outside the lockdown period.

```
Second, I believe that this study would be helped with further
exploration (or at least explanation) into the frequency
domain.  The authors say that the 0.1-50 Hz range is of
interest to volcano monitoring and contains anthropogenic
seismic noise, but don't go into further detail and should at
the very least provide more background on that choice of range
and show a spectragram for at least one station.  Dividing
that range into a few smaller ranges and processing them
individually would provide more information about the change
in the seismic noise environment (e.g.  deconvolving effects
of changing wind and water vs changing anthropogenic activity),
as well strengthen the authors' arguments regarding those noise
levels' effects on event detection.
```

We agree that more details could be provided, and we have added spectragrams and text to further differentiate what we mean with anthropogenic noise, as well as our decision to treat these data mostly in one band, as the data frequency band for the different tasks of the AVSN (ie monitoring for impending volcanic unrest, as well as seismic imaging with local seismicity) overlap with the noise sources in question.

```
As for smaller technical corrections, the main things I found
were:  the authors need to ensure that all data in a figure is
included in the one key (e.g.  figure 7's key does not contain
```

```
a red line for wind speed, and figure 8 has two keys instead
of one), decide whether to use "lock-down" or "lockdown", and
ensure that figures are more colorblind-friendly (e.g.  not
using red and green for the two different lockdown levels).
```

We have fixed the lockdown/lock-down issue, and changed to line style for level 3 to distinguish it better from level 4. Figures 7 and 8 now have two keys (each), because there are two very distinct data sets displayed in these with separate y-axes.

[Figure]

**Figure A1.** Spectrograms for the seismic data of the Auckland Volcanic Seismic Network. The vertical dashed lines indicate the start and end date of the COVID-19 lockdown in New Zealand.

**Fig. 1.**

[Figure]

**Fig. 2.**

[Figure]

**Fig. 3.**

---

## Author Comment (AC3) · 10 Dec 2020

In general, the manuscript reads well and the order of ideas
and figures is well presented.

We thank reviewer 3 for this comment.

I would like the authors to edit or re-write to sentence
starting in line 29.  I would Line 44-45:  The authors use
the frequency band:  0.1 – 50 Hz assuming the cover the range
of interest for volcano monitoring and seismic tomography,
however, whiting this range, what frequency band is the most
affected?  It would be interesting to observe the results

[Figure]

```
presented here using and plotting several frequency bands (5
or 6?  )  to understand better where the noise amplitudes show
the maximum reduction and how they are related to different
anthropogenic activities or sources (diffuse, harmonic,
transient, etc) or/and natural processes (volcanic, wind,
ocean, etc.).
```

We have rewritten this part in the revised manuscript, and added PSD plots to distinguish noise/signal in different frequency bands. What sets this study apart from previous studies on COVID-19 noise reductions is the presence of local seismicity and an active volcanic field. These require data to be used across a broad band of frequencies, where the frequencies of noise overlap with those for the signals of interest.

```
Line 73:  instead of multiplying by a factor of 25, why didn't
the authors normalized the time series presented in figure
7?  If i understand it correctly, the main idea of the figure
is to compare the relative differences between the observed
amplitudes at 3 different stations with the wind speed and more
importantly evaluate their temporal correlations.
```

Not only relative amplitudes matter and are informative; we are very much interested in the absolute value of the noise, so we can compare performance across the whole network. If we were to normalise the data in this figure, the reader loses perspective with the data from the stations not in Figure 7. Either way, the reader can appreciate how valuable it is to bury a station even by 10s of meters.

```
The fact that authors found 35 more earthquakes ( a very low
number of events) than Geonet during the lockdown, it doesn't
mean they appear because of a reduction in anthropogenic
noise, rather, they are found because of the use of a template
matching algorithm, that is, systematically more efficient in
```

[Figure]

```
finding earthquakes when compared with traditional human-based
methods.  Probably you also could find the same amount of
events even without lock down measures.  authors can add a
sentence like this after line 105.
```

We completely agree, and this was meant to be the message of this section: we find more earthquakes in general, because template matching is more sensitive than STA/LTA. We did *not* find more EQs during the lockdown than before or after the lockdown with this technique. For example, this is stated in the conclusions:

*"However, the detection rate was not higher during the lockdown period than in the periods before or after the lockdown."*

We hope the revised manuscript is more clear in this regard.
* * *
[Figure]

**Figure A1.** Spectrograms for the seismic data of the Auckland Volcanic Seismic Network. The vertical dashed lines indicate the start and end date of the COVID-19 lockdown in New Zealand.

**Fig. 1.**